# Citrus Honey Ameliorates Liver Disease and Restores Gut Microbiota in Alcohol–Feeding Mice

**DOI:** 10.3390/nu15051078

**Published:** 2023-02-21

**Authors:** Shengxiang Yi, Gaowei Zhang, Mingyan Liu, Wenjie Yu, Guohua Cheng, Liping Luo, Fangjian Ning

**Affiliations:** 1School of Life Sciences, Nanchang University, Nanchang 330031, China; 2College of Food and Health, Beijing Technology and Business University (BTBU), Beijing 100048, China; 3Agriculture and Rural Affairs Bureau of Nanfeng County, Fuzhou 344500, China

**Keywords:** citrus honey, gut microbiota, alcohol–related liver disease, short–chain fatty acids

## Abstract

Citrus honey (CH) is rich in nutrients that have a wide variety of biological functions, such as antibacterial, anti–inflammatory, and antioxidant activities, and which demonstrate therapeutic properties, such as anti–cancer and wound–healing abilities. However, the effects of CH on alcohol–related liver disease (ALD) and the intestinal microbiota remain unknown. This study aimed to determine the alleviating effects of CH on ALD and its regulatory effects on the gut microbiota in mice. In total, 26 metabolites were identified and quantified in CH, and the results suggested that the primary metabolites were abscisic acid, 3,4–dimethoxycinnamic acid, rutin, and two markers of CH, hesperetin and hesperidin. CH lowered the levels of aspartate aminotransferase, glutamate aminotransferase, and alcohol–induced hepatic edema. CH could promote the proliferation of Bacteroidetes while reducing the abundance of Firmicutes. Additionally, CH also showed some inhibitory effects on the growth of Campylobacterota and *Turicibacter*. CH enhanced the secretion of short–chain fatty acids (SCFAs), such as acetic acid, propionic acid, butyric acid, and valeric acid. Given its alleviating functions in liver tissue damage and its regulatory effects on the gut microbiota and SCFAs, CH could be a promising candidate for the therapeutic treatment of ALD.

## 1. Introduction

Alcohol consumption is currently widespread in countries around the world. The World Health Organization recently reported that about 2.3 billion people worldwide drink alcohol, and alcohol is one of the leading causes of advanced liver disease in China, Europe, and the United States [1,2]. Alcohol–related liver disease (ALD) is a degenerative disease induced by excessive alcohol consumption. ALD includes the following histopathological changes: alcoholic hepatic steatosis, alcoholic steatohepatitis, liver fibrosis, cirrhosis, and liver cancer [3]. In addition, increased oxidative stress in the liver has been recognized as one of the features of ALD [4]. Oxidative stress leads to liver damage by causing irreversible changes in lipids, proteins, and DNA and impacts biological processes [5]. For example, reactive oxygen species (ROS) produced by organelles in parenchymal cells regulate the PPARα pathway, affecting the expression of fatty acid oxidation genes in the liver [6]. Associated conditions that lead to hepatic oxidative stress include iron overload, which is associated with ROS generated by Fenton chemistry, activation of the nuclear factor erythroid–associated factor Nrf2, and the upregulation of ferritin, heme, and antioxidant components, ultimately allowing for the drastic oxidation of biomolecules, mitochondrial dysfunction, apoptosis, and/or iron toxicity [7].

The intestinal flora is diverse and closely related to the host’s health; as such, they are considered a potential method for disease treatment [8]. Liver diseases have long been associated with changes in the diversity and abundance of intestinal flora. An imbalance in the intestinal flora is considered an essential factor in the occurrence and development of ALD [9]. Alcohol causes an imbalance in the intestinal flora, increases the abundance of Gram–negative bacteria, increases the lipopolysaccharide of Gram–negative bacterial cell walls, decreases the concentration of short–chain fatty acids (SCFAs) in the intestine, and increases the permeability of the intestinal barrier [10]. SCFAs in the intestine are essential. SCFAs are dietary fiber fermentation products produced by the intestinal microbiota; they reshape intestinal ecology, induce immunomodulatory and antibiotic activity, and mediate inflammatory signaling cascades during intestinal inflammation [11].

Polyphenols are a kind of secondary metabolite composed of polyhydroxy compounds and are common in plants and their products [12]. Polyphenols are widely used in the food industry because of their antioxidant, anti–inflammatory, and anti–tumor activities [13]. In addition, polyphenols are also considered the third–largest intestinal health regulator, along with prebiotics and probiotics. Many studies have proved that the intake of polyphenols can effectively maintain intestinal microbial homeostasis and improve its structure by promoting the growth of beneficial bacteria and inhibiting harmful bacteria reproduction [14].

Citrus fruits are popular and widely grown, with a global cultivation area of over 145 million acres and an annual production of 150 million tons, accounting for one–fifth of the world’s total fruit production (FAO, https://www.fao.org/statistics/zh/ (accessed on 23 August 2022)). Citrus fruits are the fruits with the largest cultivated area and the highest yield in China. They are also an essential and stable source of honey. An average of 2050 μL of nectar can be collected from each flower via a capillary.

Citrus honey (CH) has a unique aroma and flavor and contains sugar, minerals, phenolic acids, amino acids, vitamins, and other components, meaning that it is a high–quality monofloral honey. The total phenolic content of CH is reported to be 10 to 110 mg, gallic acid equivalent 100 g^−1^, and it has a total flavonoid content of up to 10 mg, quercetin equivalent 100 g^−1^ [15]. Compared to other kinds of honey, the phenolic substances that distinguish CH are hesperetin, hesperidin, and flavanone hesperetin [16,17]. The flavanone hesperetin was considered to be the botanical marker of CH [18,19,20]. The phenolic substances in CH give it specific health–protective properties. A significant correlation between phenolic composition and antioxidant capacity was found in Italian CH [21]. Some studies have shown that CH prevents inflammation by reducing the production of the pro–inflammatory cytokine interleukin–8 and the activity of hyaluronidase and lipoxygenase and by inhibiting TNF–α–induced activation [3,22]. In summary, CH may have beneficial effects on alcoholic liver injury caused by oxidative stress and on the dysbiosis of intestinal flora and inflammatory responses [23,24]. However, to our knowledge, there are no articles about the effects of CH on ALD and the intestinal flora.

In this study, we identified the phenolic components in CH via UPLC–Q/TOF–MS. Furthermore, we explored the protective effect of phenolic substances on ALD in mice and the maintenance of microorganisms in mouse intestines. Our study may enhance the recognition of CH’s health benefits and commercial value.

## 2. Materials and Methods

### 2.1. Chemicals

Acetonitrile, methanol, ethanol, silymarin, glucose, and fructose were purchased from Sigma Aldrich (St. Louis, MO, USA), and 4–hydroxyquinoline, salicylic acid, vanillic acid, rutin, trans–4–hydroxycinnamic acid, ellagic acid, sinapinic acid, luteolin, trans–ferulic acid, hesperidin, naringin, 4–hydroxybenzoic acid, and 3,4–dimethoxycinnamic acid (purity > 98%) were purchased from Aladdin (Shanghai, China). Dihydroxybenzoic acid, methyl syringate, fisetin, abscisic acid, quercetin, trans–cinnamic acid, naringenin, apigenin, kaempferol, hesperetin, kaempferide, daidzein, and chrysin (purity > 98%) were obtained from J&K Scientific (Beijing, China). Formic acid was purchased from Merck (Darmstadt, Germany).

### 2.2. Honey

A total of five CH samples were collected by *Apis mellifera*, obtained from Shicheng County, Ganzhou City, Jiangxi Province, China. CH samples were recorded as M1–M5 respectively. All samples were tested for color, moisture, pH, acidity, conductivity, amylase values, total phenols, and total flavonoids to ensure that they met the Chinese honey standards and guaranteeing that the samples were suitable in terms of their physicochemical properties. Samples were stored individually at 4 °C and protected from light in tightly closed containers.

### 2.3. Animals

Hunan Slack Jingda Experimental Animal Co., Ltd. (Changsha, China; authorization number: (SYXK(G)2015–0002) provided 56 6–week–old male C57BL/6J mice as our experimental animal subjects. Mouse body weights (BWs) ranged between 18 and 22 g. The animals were acclimatized for one week. The temperature of the rearing environment was set at 25 ± 2 °C. The rearing environment’s light rhythm was a 12 h light/12 h dark cycle. The relative humidity of the rearing environment was 50 ± 10%. The animal study was reviewed and approved by the animal ethics review committee of Nanchang University. The date of the ethical code number was 29 December 2015.

#### Animal Experiment

Mice were randomly divided into the following groups: the pair–fed control (PF) group (distilled water, 0.05 g/Kg BW), the alcohol–fed model (AF) group (5% ethanol, 0.15 g/kg BW), the positive control (PC) group (silymarin, 0.05 g/Kg BW), the low–dose CH (LH) group (CH, 5 g/Kg BW), the medium–dose CH (MH) group (CH, 10 g/Kg BW), the high–dose CH (HH) group (CH, 20 g/Kg BW), and the fructose syrup (FG) group (fructose: glucose = 48:32, 20 g/Kg BW) [25]. Each group contained eight mice. The CH used to feed the mice was a mixture of the M1–M5 samples, with 20% of each.

After 13 weeks (91 days) of continuous group feeding, except for in the PF group, the groups were fed with 30 mL of 5% ethanol (*v*/*v*) for 10 consecutive days. On day 102, each group was treated via gavage; normal saline (5 g/Kg) was used for the PF group, and the other groups were given 31.5% ethanol (*v*/*v*). The same volume of normal saline (5 g/Kg) was fed to the PF group [26].

At the final stage of the experiment, the mice’s food intake was restricted for 12 h without affecting their water intake. The eyeballs were removed from the anesthetized mice (sodium pentobarbital, 30 mg/kg BW, i.p.) before blood (1 mL) was absorbed from the orbit, and then, EDTA–2K was injected as an anticoagulant. The mice were then sacrificed via cervical dislocation. The livers were removed and rinsed repeatedly in saline (4 °C); then, the water was absorbed using filter paper, and the livers were cut to a specific size (2 cm × 0.5 cm × 0.4 cm) and fixed with 10% formalin solution (*v*/*v*). The colon contents were placed in a sterile tube, frozen in liquid nitrogen, and stored at −80 °C.

### 2.4. UPLC–Q/TOF–MS Analysis

#### 2.4.1. Pretreatment with Honey

Before treatment, all samples were melted and homogenized at 35 °C. Zhu’s methods were used to remove sugar from the CH [27]. Then, a 1 g CH sample was mixed with 2 mL 10% NaCl hydrochloric acid solution (pH = 2) and 2 mL acetonitrile. It was vortexed for 60 s and centrifuged at 4000 rpm for 5 min, and then, the supernatant was reserved. The above steps were repeated twice; then, the supernatant was mixed and volumed at 10 mL with acetonitrile. The mixture was filtered using 0.22 μm membrane filters.

#### 2.4.2. UPLC–Q/TOF–MS Analysis

The UPLC–Q/TOF–MS analysis was performed according to the method detailed in our lab–published article (Wang et al., 2021). The samples were detected using an HSS T3 column (2.1 × 100 mm, 1.8 μm) with 0.02% formic acid water (A) and 0.02% formic acid acetonitrile (B). The primary criteria were as follows: flow rate, 0.4 mL/min; injection volume, 5 μL; column temperature, 40 °C. Linear gradient elution program is detailed in Appendix A.

### 2.5. Biochemical Analysis of Serum ALT and AST Activity

The blood was rested and centrifuged at 4 °C for 10 min at 3000 rpm to obtain the upper layer of serum. The bioactivities of serum alanine aminotransaminase (ALT) and aspartate aminotransferase (AST) were measured with a colorimetric procedure using detection kits (Nanjing Jiancheng Biotechnology, Nanjing, China) according to the manufacturer’s protocol.

### 2.6. Histopathological Analysis of the Liver

The liver was fixed by soaking for 48 h in a 4% paraformaldehyde solution (*v*/*v*). It was then dehydrated with a series of graded ethanol solutions, embedded in paraffin, sliced (4 μm), and stained with hematoxylin and eosin (H&E). A microscope imaging system captured the histological photographs (Leica DM1000, Nussloch, Germany).

### 2.7. Gut Microbiota Analysis

The E.Z.N.A. soil DNA Kit (Omega Bio–tek, Norcross, GA, US) was used to isolate total DNA from mouse feces, and quantitative and qualitative evaluations were conducted to evaluate eligibility for future sequencing. The chosen primer pairs were 338F (5′–ACTCCTACGGGAGGCAGCAG–3′) and 806R (5′–GGACTACHVGGGTWTCTAAT–3′). The V3–V4 region of the bacterial small subunit 16S rRNA gene was amplified and quantified using a PCR technique to create a sequencing library. Using the Illumina HisSeq platform (Illumina, San Diego City, CA, USA), the PCR–amplified products were then sequenced. The raw reads were submitted to NCBI’s Sequence Read Archive (SRA) database.

### 2.8. Processing of Sequencing Data

According to the following criteria, the raw 16S Rrna gene sequencing reads were demultiplexed, quality–filtered using Trimmomatic, and merged using FLASH: only overlapping sequences longer than 10 bp were constructed in accordance with their overlapped sequence, and 300 bp reads were cut at any place with an average quality score of “discarded”. The overlap region’s maximum mismatch ratio was 0.2. Reads that could not be put together were discarded, and samples were identified using barcodes and primers. For exact barcode matching and primer matching with two mismatching nucleotides, the sequence orientation was changed.

UPARSE (version 7.1, http://drive5. Com/uparse/ (accessed on 15 June 2022)) was used to cluster operational taxonomic units (OTUs) with a 97% similarity criterion, and chimeric sequences were found and eliminated. The Ribosomal Database Project (RDP) Classifier (http://rdp.cme.msu.edu/ (accessed on 15 June 2022)) was used to compare the taxonomy of each OTU representative sequence to the 16S rRNA database (e.g., Silva SSU128) using a confidence level of 0.7.

### 2.9. Determination of SCFAs in the Colon

Target SCFAs in mouse feces include acetic acid, propionic acid, butyric acid, and valeric acid. Determination was performed via gas chromatography in combination with mass spectrometry (GC–MS). Before analysis, the samples were derived according to Gu’s method [28]. To determine the concentration levels of SCFAs in the mouse colon, contents of three colons were randomly selected from each group to detect SCFAs, and the SCFAs were qualitatively quantified using the standard curve equation produced from standards to derive the concentration of SCFAs in each sample. The analytical conditions for gas chromatography were as follows: an HP–5MS column (30 m × 250 μm × 0.25 μm) with a sample inlet of 250 °C; helium was the carrier gas (purity ≥ 99.999%) with a flow rate of 1 mL/min; shunt ratio of 10:1; program heating, 60 °C maintained for 2 min, raised by 6 °C/min to 200 °C, and held for 0 min; injection volume of 1 μL; the solvent was delayed by 5 min. The mass spectrometry conditions were as follows: EI ion source of 230 °C; four–stage rod of 150 °C; 70 eV electronic collision mode, sample acquisition at a scanning range of 50–600 *m*/*z* to obtain mass spectrometry; run for 2 min after reaching 300 °C. In SIM mode, the peak area of the sample was screened according to the mass–to–charge ratio, such as 117 acetic acid–to–charge ratios. The standard curve was generated and is shown in Appendix A.

### 2.10. Statistical Analysis

Data were exported to Microsoft Excel and expressed as the mean ± standard deviation (SD). IBM SPSS Statistics 25 (Armonk, NewYork, NY, USA) was used to perform a one–way ANOVA and Tukey’s test to determine the significance of differences, with *p* < 0.05 indicating statistical significance.

## 3. Results

### 3.1. Analysis of Compounds in CH Using UPLC–Q/TOF–MS

A total of 22 chemical components were identified in negative ion mode and 19 in the positive ion mode. Combining the results, a total of 41 chemical components were identified in CH, mainly comprising flavonoids, phenolic acids, alkaloids, and terpenoids. A total of 18 chemical components were detected in all samples; these primarily consisted of terpenoids (abscisic acid), alkaloids (4–hydroxyquinoline), phenolic acids (3,4–dihydroxybenzoic acid, 4–hydroxybenzoic acid, ellagic acid, and 2, 5–dihydroxybenzoic acid), and flavonoids (3,4–dimethoxycinnamic acid, rutin, naringenin, kaempferol, chrysin, etc.). Among the 41 compounds, 26 were quantified by comparing the retention time and mass spectrometry information with existing standards, and the results are shown in Appendix A. The BPI of CH is shown in Appendix A.

### 3.2. Influence of CH on Serum ALT and AST Activity

According to Figure 1A, the values of AST and ALT in mouse serum were relatively consistent across the differential experimental groups. First, the two paraments presented the lowest values in the PF group. AF showed an increment of both values, while PC exhibited the opposite result to that of AF. Accordingly, the variation in the AST and ALT values reflected the alleviating effect of drugs on alcoholic liver injury. Second, The AST and ALT values were reduced in the LH and HH groups compared to those in the AF and FG groups. Overall, honey was shown to have a particular effect on mitigating the syndromes triggered by alcohol, on a basis related to the intake dose. Studies have shown that multifloral honey can suppress the increase in ALT and AST values due to acute alcohol intake [29].

### 3.3. Histological Examination of Liver Sections

As depicted in Figure 1B, the liver cells of the PF group possessed a complete morphology, an evident nucleus, and a cell boundary, and the hepatic cords were distributed in an orderly fashion inside the tissue. In the AF group, the liver cords were disorganized and indistinct, as were the margins of hepatocytes and the cell nucleoplasm. The liver cell damage caused by alcohol in mice was improved by honey intake, and cellular edema and steatosis were reduced to a lower level. Compared with the AF group, the honey group showed that honey had a protective effect on liver tissue under the influence of alcohol. Quantitative scoring results for the proportion of steatosis, inflammation, and tissue fibrosis in liver tissue are presented in Appendix A.

### 3.4. Influence of CH on Gut Microbiota

Dilution curves for gut microbiota sequencing were constructed with randomly selected samples and calculations of their numbers and representative OTUs. A threshold of 97% for similarity in OTUs was chosen in this study. The trend of the dilution curve indicated that the amount of sequencing data examined in this study was sufficient, and the sequencing depth covered all species of the samples. The Shannon–Wiener index curve showed a flat sample curve, indicating that the sample species richness was good at the current sequencing depth and could meet the requirements for further analysis. The highest values among the samples showed differences in microbial diversity. The Chao indices, ACE indices, Shannon indices, and Simpson indices can be used to illustrate alpha diversity, which is related to bacterial diversity in a particular region or environment. The Chao and ACE are positively correlated with species abundance. The larger the Shannon index, the lower the species diversity, while the higher the Simpson index, the lower the species diversity (Appendix A).

According to Appendix A, the species abundance and diversity of the gut microbiota did not differ significantly between the groups. The Chao and ACE values indicated that the species abundance of other groups increased compared with that in the PF group. The Shannon and Simpson indices showed that gut microbiota diversity decreased in the honey groups, and especially in the MH group.

Figure 2A,B and Appendix A show the species proportion of gut microbiota in each group of mice at the phylum level. The species with a high balance of gut microbiota comprised Proteobacteria, Bacteroidota, Actinobacteriota, Firmicutes, Campylobacterota, Verrucomicrobiota, and Desulfobacterota. In contrast with that in the PF group, the proportion of Bacteroidota in the AF group decreased, but the ratio of Verrucomicrobiota, Firmicutes, Proteobacteria, Actinobacteriota, and Desulfobacterota increased. Gut flora composition was compared between the PC and PF groups with no significant variation. The proportion of Verrucomicrobiota increased slightly and more in the PC group than in the AF group. In Celiberto’s study, chronic ethanol intake was shown to significantly change the balance of the gut microbiota, increasing the F/B ratio in particular [30]. Using the AF group as a reference, the F/B ratio showed a consistent decreasing trend in the honey and PC groups, but the effect of the honey differed according to different concentration settings. However, the ability of the LH group to alleviate alcohol–induced gut microbiota imbalances tended to be consistent with that of the PC group. Moreover, the honey–treated mice showed stronger resilience to gut microbiota imbalances than the FG group; this was consistent with the findings of liver pathology tests, which indicated a reduction in alcohol–induced liver damage. Specifically, LH and HH could maintain the abundance of Bacteroidota and reduce that of Firmicutes, Actinobacteria, and Verrucomicrobiota. With the increase in honey absorption, the abundance of Proteobacteria was depressed.

Principal coordinates analysis (PCoA) does not influence the positional relationship between sample locations, but it does modify the coordinate system to examine the similarities between data. The differences between individuals or groups can be observed using PCoA. Figure 2C shows that independent clusters made up the constitution of the gut microbes in each experimental group. AF was relatively far away from the other groups and did not overlap. This result demonstrated that alcohol alters the diversity of intestinal microbial species. The distance between the remaining groups was relatively small, except for that of MH, which was somewhat scattered. The honey groups maintained the same level of gut microbiota composition as PC and PF, with the exception of MH.

Hierarchical cluster analysis was carried out on all samples using a Bray–Curtis algorithm. According to the similarity between the samples, a tree diagram was created to represent the composition of the sample’s bacterial community. The position of different samples on the tree shows their similarity and intuitively exhibits the influence of each treatment method on the composition level of gut bacteria. Figure 2D demonstrates a significant variance in bacterial organization among the AF, FG, and other treatment groups; meanwhile, the PF, PC, LH, and HH groups had similar bacterial compositions. This corroborates the results of the PCoA. It was demonstrated that honey is not only composed of sugars but also contains trace substances that have a mitigating effect on alcohol–induced intestinal flora dysbiosis.

To investigate further the influence of honey on the gut microbiota of mice fed alcohol, we identified precisely the altered bacterial phenotypes using LEfSe analysis. As shown in Figure 3, the differential microbial taxa for different treatments were obtained using LDA > 4.0 as the screening condition. As shown in Figure 3A, the differentiated microbial taxa in PF were mainly composed of the phylum Bacteroidota, family Muribaculaceae, genus Faecalibaculum, and their subordinate microorganisms. The taxa of AF were primarily composed of Firmicutes, Erysipelotrichales, *Turicibacter*, Clostridia, and *norank_f_norank_o__Clostridia_UCG–014*. The taxon of PC only consisted of Prevotellaceae and its subordinate microorganisms, such as the *uncultured_bacterium_g__Prevotellaceae_NK3B31_group*. The taxa of FG were *Clostridium_sensu_stricto_1*, Clostridiaceae, and Clostridiales. As shown in Figure 3B, differential microorganisms in the AF group included Firmicutes, Erysipelotrichales, *Turicibacter,* and Clostridia, etc., in addition to Actinobacteriota and Bifidobacteriales. The differential marker microorganisms of MH were mainly composed of the order Lactobacillales and its subordinate microbiota. The differential marker microorganisms of HH were primarily composed of the genus Faecalibaculum and its subordinate microbiota.

### 3.5. Effect of CH on Intestinal Microbial Function

The metabolic activity of the gut microbiota in alcohol–fed mice was predicted using PICRUSt analysis based on the KEGG and COG databases. It can be seen from Figure 4A,B that under the influence of alcohol, the total metabolic degree of the AF group was the highest compared with that of the other experimental groups and included ribosomes related to phagocyte defects, ABC transporters associated with hyperbilirubinemia, DNA replication associated with immunodeficiency, and homologous recombination related to defects in RecQ helicases.

### 3.6. Correlation Analysis of SCFAs and Gut Microbiota

Alcohol treatment elevated the metabolism of acetic acid in mouse feces (Figure 5). CH treatments inhibited that promoting effect, and the inhibitory effect of MH and HH was more prominent. Feeding with alcohol decreased the concentrations of propionic, butyric, and valeric acids in mouse feces. However, CH could alleviate the above–mentioned reduction in SCFAs caused by ethanol, and the effect of low–dose CH was more effective.

The gut microbiota mainly produce SCFAs in the colon. An analysis of the correlations of SCFAs, the gut microbial composition, and the relative gut microbial abundance is shown in Figure 6. Butyric acid was significantly and positively correlated with Patescibacteria; acetic acid was significantly and negatively associated with Cyanobacteria. At the family level, butyric acid was strongly correlated with Saccharimonadaceae. Valeric acid had a negative connection with Turicibacter at the genus level.

## 4. Discussion

CH production is widely distributed; the product is rich in nutrients and bioactive ingredients, which are popular among consumers. However, its pharmacological effects still need to be investigated. By clarifying the chemical composition of CH and conducting experiments in mice, the present study provides preliminary evidence that CH alleviates the progression of ALD and regulates the intestinal microbiota.

First, the metabolite in CH with the highest concentration was abscisic acid, with a range of 235.56–276.20 μg/100 g. It has been shown that abscisic acid is a novel PPARγ agonist that improves insulin resistance and suppresses systemic inflammation in obese mice [31]. The detected contents of 3,4–dimethoxycinnamic acid ranged from 89.60 μg/100 g to 111.27 μg/100 g. The biological activities of 3,4–dimethoxycinnamic acid are mainly involved in reducing cholesterol [32,33]. The rutin content in CH was 62.88–81.66 μg/100 g. Rutin is a common flavonoid with antibacterial, anti–inflammatory, anticancer, and hypoglycemic pharmacological properties, among others [34]. Hesperetin and hesperidin were markers of CH. Hesperetin and its derivatives have anticancer, antioxidant, anti–inflammatory, anti–hyperglycemic, and anti–hypertensive properties [33,35,36]. Hesperidin alleviates alcoholic liver steatosis caused by alcohol stimulation in mice [37,38]. Clarifying the phenolic composition of CH improves our understanding of the potential mechanisms underpinning the effects of CH in the designed animal experiments.

Second, hepatocyte lipid peroxidation is the main mechanism of liver injury after chronic alcohol consumption, and the lipid peroxidation of phospholipids increases hepatocyte permeability, which can lead to the leakage of cellular enzymes into the blood [39]. Liver pathology sections further demonstrated the protective effects of CH on hepatocytes and liver tissue. The reason for the absence of significant differences between groups may be due to individual mouse differences and sample loss due to experimental manipulation during serum collection, which ultimately resulted in a large standard deviation of the data. However, from the results, it can be concluded that CH has some lowering effect on AST and ALT, which was consistent with the therapeutic effect of other kinds of honey on ALD. Other studies have used jujube honey as an experimental material to demonstrate the down–regulation of ALT and AST mediated by honey [40]. In the pathological analysis of liver sections, CH resulted in clearer hepatocyte boundaries, mainly in the morphology of the cells surrounding the central vein. The general nucleus was apparent under CH intervention. This result was highly consistent with Luo’s findings [41]. Our analysis found that the remission effect on ALD seen in the LH group and HH groups was better than that observed in the MH group. Other than the fact that the maintenance of hepatic cord tissue in the LH group was weaker than that in the PC group, the other effects are similar to those observed in the PC group.

Third, the heatmap, PCoA, hierarchical clustering tree, and bar plot all reflected the alterations in the intestinal microbiota in AF, and the combination of the results for LH, MH, and HH showed that CH can, to some extent, control the alteration of the intestinal microbiota caused by alcohol intake. Combined with the intestinal microbiota data of the FG group, this finding indicates that CH affects the intestinal microbiota independent of its sugar composition. However, it was worth noting that the CH treatment did not show the dose effect that it should have. It is speculated that the reason for this result is that individual mice differ in their absorption of CH or that the dose gradient of CH was not set precisely enough. Further LEfSe analysis demonstrated statistically different differential intestinal microbiota between the groups. The differential microorganisms in both the AF and PC groups consisted of common microbiota in the intestinal tract. However, the AF and FG groups showed that potentially pathogenic microorganisms, such as norank_f__norank_o__Clostridia_UCG–014 in the AF group, might be associated with ulcerative colitis [42]. Costridiaceae in the FG group may be associated with diabetes mellitus [43]. The results of the comparative LEfSe analysis of the AF and CH treatment groups showed that more differential flora appeared in the AF group due to the influence of alcohol. The class Bacilli was the most significant microbiota taxon in the AF group, which may correlate with the elevation of the genus Bacillus due to alcohol consumption, which has been found in other studies [44]. Erysipelotrichales was also a differential gut microorganism taxon found in the AF group; in other studies, alcohol intake was found to lead to changes in the composition of the small intestinal microbiota, and the relative abundance of Erysipelotrichales was dominant, which is consistent with our findings [45]. The presence of Turicibacter in AF is not easy to ignore, as a correlation between Turicibacter and alcoholic liver damage has been found in other studies [46]. In contrast, Bacilli, Erysipelotrichales, and Turicibacter were not found in the CH–treated group under the influence of alcohol, which may indicate, to some extent, the effect of CH on alcohol–induced dysbiosis of the intestinal microbiota. It is noteworthy that the LH group did not show differential microorganisms when the LEfSe analysis was performed, probably because the lower CH dose had less of an effect on intestinal microorganisms, further reflecting the need for refinement in the CH dose setting of the experiment. The differential microorganism of MH was Lactobacillus_johnsonii, a probiotic that has been reported to have a potential role in modulating immunotherapy [47]. Faecalibaculum_rodentium in HH has potential antitumor effects [48]. In summary, alcohol causes disorders in the intestinal microbiota and produces some harmful microorganisms, while the intervention of CH promotes the growth and colonization of some probiotics, thus contributing, to a certain extent, to the maintenance of the homeostasis of the intestinal microbiota.

Fourth, alcohol interferes with several metabolic processes, including protein metabolism, ribosomal structure, chromatin structure, protein conversion, nucleotide transport, and metabolism. The effect of low and high doses of CH on metabolic pathways related to the intestinal flora is opposite to the effect of alcohol on those pathways. Medium doses of CH were less effective in alleviating the above–mentioned disorders of intestinal flora metabolism caused by alcohol. The intestinal flora under the influence of FG exacerbates the associated metabolic disturbance. ABC transporters are increased under the influence of alcohol, and ABC transporters are related to hyperbilirubinemia [49]. Honey could reduce the level of ABC transporters, an effect that was particularly pronounced for the LH and HH groups, which is consistent with the previous conclusions.

Fifth, intestinal microbiota in the intestine produce SCFAs. Recent studies have shown that SCFAs can have many beneficial effects on ALD in a variety of ways [50]. According to the results of SCFAs, alcohol intake increased the acetic acid content and decreased the content of other SCFAs in the colon. It has been demonstrated that mammalian intestinal metabolites are altered under the long–term influence of ethanol, and the effect on SCFAs was mainly reflected by a significant reduction in unexpected SCFAs, except for acetic acid; this finding is consistent with the results of this experiment [51]. It was found that propionic acid in the intestine intervenes in autoimmune prostatitis, reduces susceptibility to autoimmune prostatitis, and corrects the imbalance in Th17/Treg cell differentiation in vitro and in vivo [52]. The primary nutrient used for intestinal cells is butyric acid, which also controls the expression of genes by inhibiting histone deacetylase [53]. Zhao’s study found that the ingestion of polydatin, a natural product, promoted elevated fecal valeric and hexanoic acids in mice, which in turn activated the AMPK signaling pathway [54]. In this experiment, CH treatment slightly increased the contents of propionic acid, butyric acid, and valeric acid. Moreover, alcohol caused a decrease in the content of the above SCFAs. It is worth mentioning that the levels of propionic acid, butyric acid, and valeric acid were lower in the PC group than in the AF group. It is speculated that the intake of silymarin would alleviate the liver damage caused by alcohol but have a negative effect on the production of SCFAs, reflecting the fact that CH treatment has some advantages compared to drugs. Statistically, the data of SCFAs under honey treatment were not significantly different from each group. However, the results exhibit that the intake of honey leads to a decrease in the level of acetic acid. Also under the influence of honey, intestinal flora will maintain or even increase the metabolism of other SCFAs. The results of the correlation analysis between SCFAs and the intestinal flora may demonstrate how phenolics in CH may alter the metabolism of the intestinal flora, thus further alleviating ALD. Our results demonstrated a significant positive correlation between Patescibacteria and butyric acid, a pair that is beneficial for the gut. This may provide some support for Nie’s finding that the gut microbes that contribute most to fecal microbial diversity in mice treated with herbal remedies for colon cancer contain Patescibacteria [55]. Butyric acid showed a significant positive correlation with Saccharimonadaceae, and some studies have shown that Saccharimonadaceae is positively correlated with the immune response [56].

Finally, the present experiment further investigated the interventional effects of CH on ALD based on the clarification of the chemical composition of CH, the main effects of which were the alleviation of liver cell and tissue damage and the improvement of intestinal flora disorders caused by alcohol. However, this experiment has many shortcomings; for example, the experimental results show that the experimental dose of CH should be set using a greater number of gradients, so as to describe the dose effect of CH more accurately. The effect of CH on the improvement of the intestinal flora was not verified through fecal transplantation. These shortcomings indicate directions for future experiments.

## 5. Conclusions

UPLC–Q/TOF–MS analysis revealed that the chemicals in CH were mainly composed of biologically active phenolics, such as abscisic acid, 3,4–dimethoxycinnamic acid, rutin, hesperetin, and hesperidin. CH showed some substantial benefits in decreasing ALT and AST values, maintaining the homeostasis of the gut microbiota, and manipulating the gut microbiota to control the production of SCFAs. The increasing impact of honey changes according to the dose administered. As a rich mixture of active ingredients, honey may attenuate the adverse effects caused by chronic or excessive alcohol consumption through different metabolic pathways, with varying thresholds and interactions between the underlying molecules. Changes in honey composition may alter the mechanisms and pathways by which cells or microorganisms respond to ethanol, resulting in different therapeutic effects for different types of honey. Therefore, further experiments should be conducted to investigate the regulatory mechanisms and mechanisms of action of various CH metabolites in the liver, as well as with respect to the intestinal flora, in order to address diseases associated with acute or chronic alcohol damage.

## Figures and Tables

**Figure 1 nutrients-15-01078-f001:**
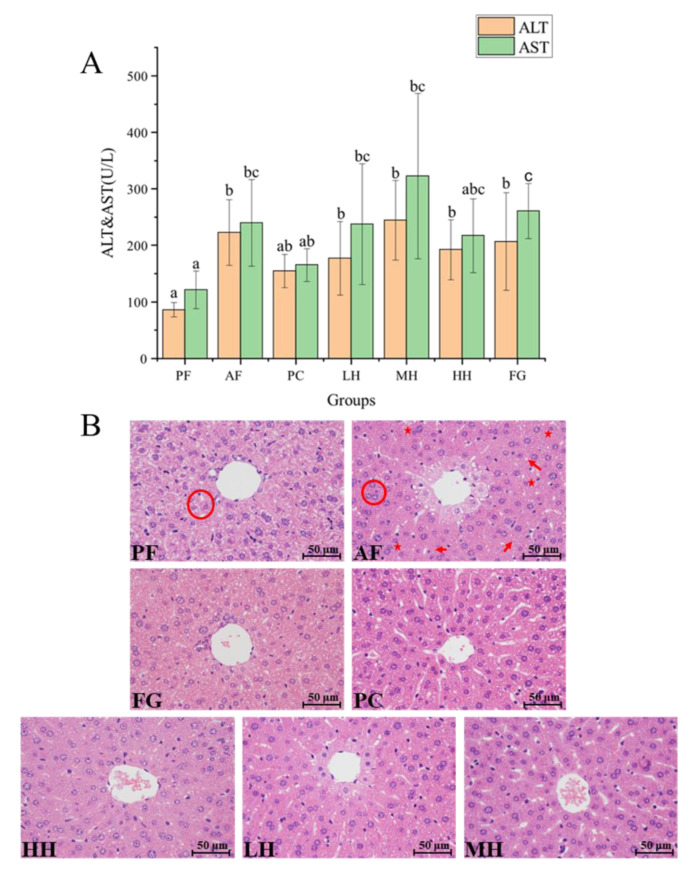
(**A**) Serum alanine aminotransferase (ALT) and aspartate aminotransferase (AST) levels of mice in each group. Different letters a, b, and c represent the significant differences between groups (*p* < 0.05). (**B**) Liver pathology sections stained with hematoxylin and eosin (HE × 400). Highlights are the disappearance of a cell gap (encircled), inflammation (stars), and hepatic steatosis (bold arrows). Note: PF represents the pair–fed control group. AF represents the alcohol–fed model group. PC represents the positive control group. LH represents the low–dose citrus honey group. MH represents the medium–dose citrus honey group. HH represents the high–dose citrus honey group. FG represents the fructose syrup group.

**Figure 2 nutrients-15-01078-f002:**
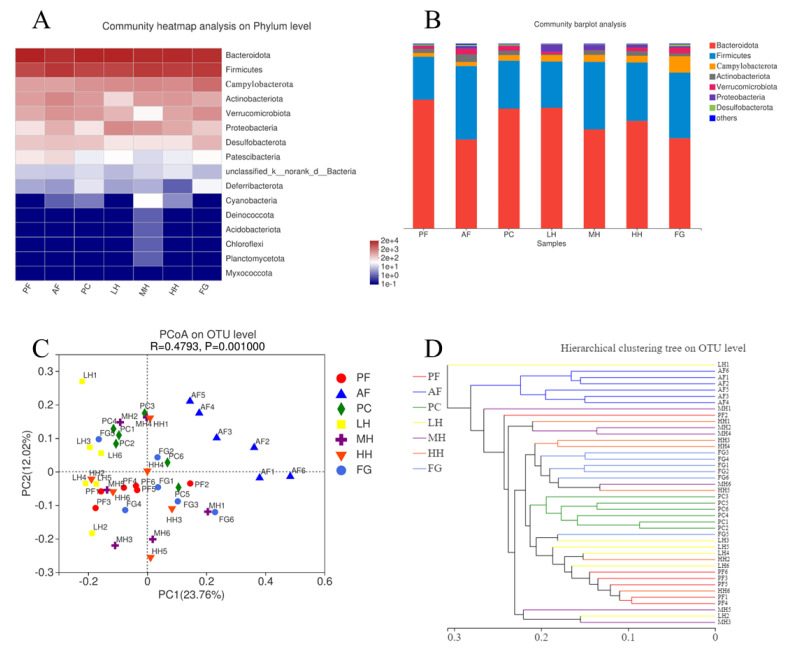
Citrus honey alters the gut microbiota structure in mice. Structures of gut microbiota of mice were analyzed using 16S rRNA gene sequencing and bioinformatics analysis. (**A**) Relative analysis of the community heatmap at the phylum level. (**B**) Composition and relative abundances of the gut microbiota at the phylum level. (**C**) Unweighted uniFrac–based principal coordinates analysis (PCoA) plots. (**D**) Operational taxonomic units (OTU) level hierarchical clustering tree. Note: PF represents the pair–fed control group. AF represents the alcohol–fed model group. PC represents the positive control group. LH represents the low–dose citrus honey group. MH represents the medium–dose citrus honey group. HH represents the high–dose citrus honey group. FG represents the fructose syrup group.

**Figure 3 nutrients-15-01078-f003:**
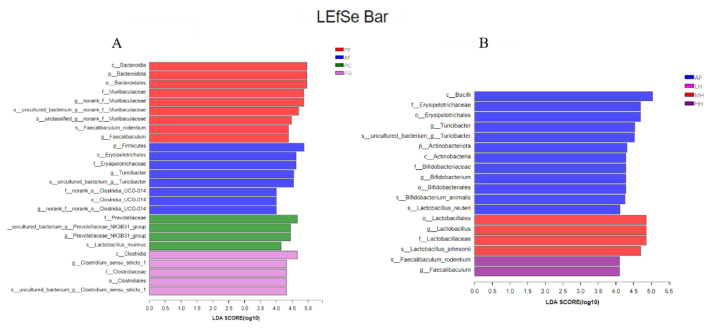
Dominant taxa in the mouse gut microbiota, with a linear discriminant analysis (LDA) score > 4.0, *p* < 0.05. (**A**) Comparative linear discriminant analysis Effect Size (LEfSe) analysis of mice in the PF, AF, PC, and FG groups (**B**) Comparative LEfSe analysis of mice in the AF, LH, MH, and HH groups. Note: p: phylum, c: class, o order, f: family, g: genus; different colors indicate different groupings. PF represents the pair–fed control group. AF represents the alcohol–fed model group. PC represents the positive control group. LH represents the low–dose citrus honey group. MH represents the medium–dose citrus honey group. HH represents the high–dose citrus honey group. FG represents the fructose syrup group.

**Figure 4 nutrients-15-01078-f004:**
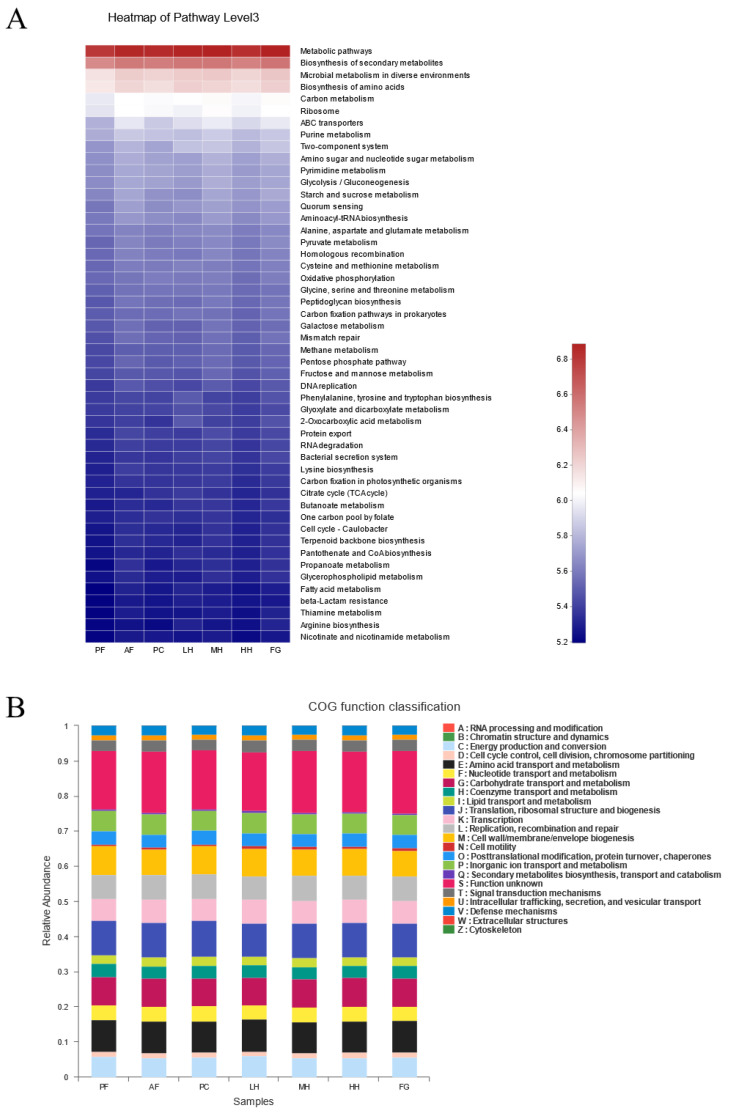
(**A**) Heatmap of major pathways in the KEGG database at level 3, based on PICRUSt anticipated functions of the gut microbiota; the colors span from blue to red. (**B**) The relative abundances of gut microbiota in the classification of the COG function. Note: PF represents the pair–fed control group. AF represents the alcohol–fed model group. PC represents the positive control group. LH represents the low–dose citrus honey group. MH represents the medium–dose citrus honey group. HH represents the high–dose citrus honey group. FG represents the fructose syrup group.

**Figure 5 nutrients-15-01078-f005:**
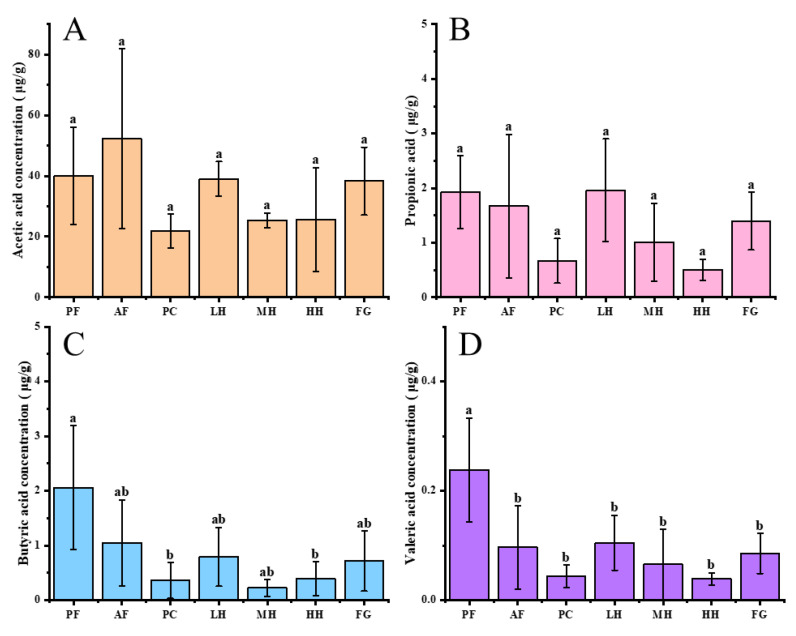
Concentration of SCFAs in the colonic contents of mice in each group. (**A**) Acetic acid. (**B**) Propionic acid. (**C**) Butyric acid. (**D**) Valeric acid. Different letters a and b represent the significant difference between groups (*p* < 0.05). Note: PF represents the pair–fed control group. AF represents the alcohol–fed model group. PC represents the positive control group. LH represents the low–dose citrus honey group. MH represents the medium–dose citrus honey group. HH represents the high–dose citrus honey group. FG represents the fructose syrup group.

**Figure 6 nutrients-15-01078-f006:**
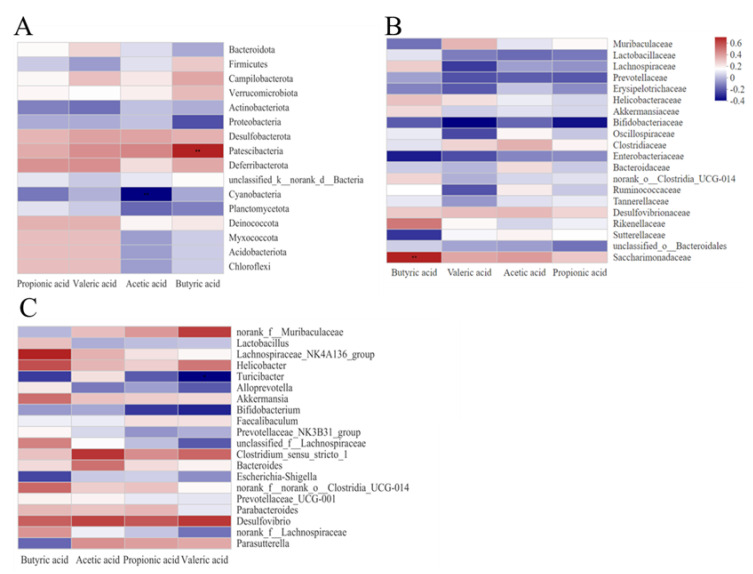
Cluster heat map for correlation analysis of acetic acid, propionic acid, butyric acid, and valeric acid with gut microbiota in mouse colonic contents. (**A**) Cluster heat map for the correlation analysis of acetic acid, propionic acid, butyric acid, and valeric acid with gut flora at the phylum level. (**B**) Cluster heat map for correlation analysis of acetic acid, propionic acid, butyric acid, and valeric acid with gut microbiota at the family level. (**C**) Cluster heat map of correlation analysis of acetic acid, propionic acid, butyric acid, and valeric acid with gut microbiota at the genus level. Note: red represents a positive correlation, blue represents a negative correlation, ** *p* < 0.01. Note: PF represents the pair–fed control group. AF represents the alcohol–fed model group. PC represents the positive control group. LH represents the low–dose citrus honey group. MH represents the medium–dose citrus honey group. HH represents the high–dose citrus honey group. FG represents the fructose syrup group.

## Data Availability

The data that support the findings of this study are available from the corresponding author upon reasonable request.

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
