# Peer review of "Citrus Honey Ameliorates Liver Disease and Restores Gut Microbiota in Alcohol–Feeding Mice"

_nutrients, 2023, doi:10.3390/nu15051078_

Round 1

Reviewer 1 Report

The manuscript is very interesting. The methodology used is updated and sufficient. The results support the discussion. However, I have the following comments.

I. Major comments:

1. In the introduction I suggest including a brief paragraph regarding the adverse effects of alcohol (liver but also on public health)

2. A relevant aspect in the development of liver damage is the increase in oxidative stress. include a brief paragraph on this point in the introduction.

3. The beneficial effects of Citrus Honey were probably generated by the antioxidants it contains. In this regard, would it be possible for the authors to carry out experiments to assess changes in oxidative stress and the antioxidant response (especially in the liver)?

4. The introduction is good, but it is necessary that the authors discuss molecular or mechanistic aspects that allow a better understanding of the results.

4.1. Briefly discuss the role of oxidative stress in liver damage, considering possible pathways involved (inflammation, mitochondrial dysfunction, fatty acid synthesis, etc.) PMID: 34687092

II. Minor comments:

1. Improve the wording of the objective of the study.

2. It would be interesting if the authors could include a figure summarizing the main effects observed.

Reviewer 2 Report

General comment: The main problem with the article is the English language which should be polished as it is difficult to understand, so the conclusions the authors want to present are unclear.

Specific comments:

The abstract

Line 16, 17 Duplicated sentence „There were 26 metabolites identified 16 and quantified in CH in total. There were 26 metabolites identified and quantified in CH…”

Line 22 Campylobacterota are a phylum of bacteria- not „Campilobacterota” the term should be corrected

The introduction includes a mixture of different data not connected smoothly to each other

Line 31-32 „The production of ALD can lead to other liver pathologies, such as steatohepatitis, 31 alcoholic hepatitis, liver fibrosis, cirrhosis, and liver cancer” ??? what does it mean „the production of ALD”

Line 34 „damage to inflammatory mediators” ???

Line 37 the intestinal flora has exceeded 1014 ???

Line 55 Citrus ??? is one of the most important fruits

Line 70- The description of citrus honey is finished with a different subject

Line 70-75 what inflammatory process do the authors mean?

Material and methods

Line 115-116 the other groups were fed with 30 mL 115 of 5% ethanol (v/v) for ten consecutive days at the 13th week. On the next eleventh day…???

Statistical analysis

Line 194 All data were from more than three repeated times ???

Figure 1 Varied letters indicate different significance (P < 0.05)- what does it mean?

Figures All abbreviations used should be explained to make the figure self-explanatory.

Line 223

3.3. Liver histopathological of alcoholic-induced liver damage ???

Discussion and conclusions

The authors use the term alcohol impairment which is too general and should be specified.

Institutional Review Board Statement: there is no data

My conclusion: a major revision with extensive editing of the English language and style is required. 

Reviewer 3 Report

This study is very important because it examines the effects of citrus honey on gut microbiota in ALD. Citrus honey has 26 metabolites with a wide variety of biological functions. However, the effects of CH on ALD and gut microbiota are not clear. This study aimed to determine CH's alleviation effects on ALD and CH's regulatory effects on gut microbiota in mice. The animal study was reviewed and approved by the animal Ethics review committee of Nanchang University.

This study provides preliminary evidence that CH alleviates the progression of ALD and regulates gut microbiota by clarifying the chemical composition of CH and experiments in mice.

Minor Comments:

2.5. Biochemical analysis of serum ALT and AST activity (add) (since only these parameters are determined)

3. Results

3.2. Influence of CH on serum physiological indicators  ALT and AST activity

In Figure 1. mark the keys histopathological changes of hepatocytes in ALD  (star, arrow..)

Figures 2-6 are very clear, that is, the results are excellently presented in these Figures.

Round 2

Reviewer 1 Report

Authors answered all my comments. Therefore, the manuscript can be accepted. 

Author Response

Thank you again for your review of our manuscript. It was our pleasure to work with you. Thank you for your professional work and for giving us the opportunity to have our manuscript accepted!

Reviewer 2 Report

The Authors have not paid attention to all my previous comments. There are still numerous issues to be addressed. The mistake Campilobacterota" is still included in the text of the manuscript. According to the current recommendations, the term alcoholic liver disease should be modified into alcohol-related liver disease. Line 52 "the negative bacteria’s cell walls" - should be Gram-negative. The content of the following subsections 2.1. Chemicals and 2.2. Honey has been duplicated. English has been polished, but still, there are mistakes that need correction. Other specific comments are included in the attached file
